# Particle and Particle Agglomerate Size Monitoring by Scanning Probe Microscope

Pavel Gulyaev [1], Tibor Krenicky [2], Evgeny Shelkovnikov [1] and Aleksandr Korshunov [1,*]

[1] Udmurt Federal Research Centre of the Ural Branch of the Russian Academy of Sciences, Institute of Mechanics, 426067 Izhevsk, Russia; lucac@e-izhevsk.ru (P.G.); iit@udman.ru (E.S.)
[2] Department of Technical Systems Design and Monitoring, Faculty of Manufacturing Technologies with a Seat in Presov, Technical University of Kosice, Sturova 31, 080 01 Presov, Slovakia; tibor.krenicky@tuke.sk
[*] Correspondence: kai@udman.ru

**Featured Application:** The results presented in this paper can be used in microscopy and modern methods of particle inspection using images.

**Abstract:** In the present study, the use of a scanning probe microscope is described for monitoring the sizes of nanoparticles. Monitoring is the process of acquiring and analysing the set of overlapping images. The main analysis steps are image segmentation, determination of nanoparticles allocation and their sizes, determination of the overlap of images with one another, and exclusion of repeating measurements for the formation of the correct particle-size sampling. The thorough examination of commercial scanning probe microscopes, software, and image processing libraries showed that their capabilities are limited for image segmentation and determination of sizes in complex structured images. A method based on the surface curvature computation is proposed for the image segmentation (allocation of particles) and determination of particle sizes. The curvature is estimated using the surface area approximation with respect to the circumference. It is proposed to use sample displacement sensors as an aid for image stitching.

**Keywords:** scanning probe microscope; particle size monitoring; key points; descriptor; image stitching; segmentation; particle allocation

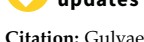



## 1. Introduction

The physical and chemical properties of nanoparticles are mainly determined by their sizes. Size monitoring is an important component of the process of nanoparticles production [1,2]. The monitoring is the formation of the particle-size sampling followed by the analysis of the particle-size distribution. The analysis is of importance for the evaluation of the sizes of particles themselves and for the evaluation of particle-based products, for example, of the dispersity evaluation of powders containing particles under study. For the data acquisition and formation of the particle-size distribution such methods as laser diffraction [3,4], dynamic light scattering [5,6] and acoustic spectroscopy [7,8] are more widely used. The methods are mainly suitable for controlling particles in solutions. The disadvantage of these methods is measurement results distortion, which can occur with a significant difference in the particles size present in the solution.

In this study, we consider the case when nanoparticles are deposited on a substrate before research and size monitoring, carried out by images. An overview of deposition methods and a description of the simplest one can be found in [9]. Deposition is widely used for particles obtained in the liquid phase and in the form of powders. In the latter case, a suspension or colloidal solution is prepared from the particles. Traditional photography [10] or various microscopes and diffractometers [11–17] are widely used to obtain particles images on substrates. One of the most powerful tools among them is a scanning probe microscope (SPM). In comparison with other devices such as an electron microscope, for

example, the SPM has the following advantages: variety of sample types, functionality in air and liquid, relative simplicity of sample preparation, direct topographic measurements and the presence of nanolithography functions permitting to mark studied regions during monitoring. Since the lifetime of particles on the substrate can be limited, and the execution time of one scan can reach 5–10 min, significant efforts of developers are aimed at time reduction and automating the monitoring of particle sizes. One of the ways to reduce monitoring time is to substitute sequential uniform scanning [18] with tracking scanning. Automation is possible due to additional image processing and the determination of tracking scan parameters based on the results of this processing.

Direct topographic measurements in SPM allow to use the conventional image processing methods. As a rule, SPMimages are processed with the use of conventional algorithms [19] presented in OpenCV and FLANN libraries [20–22]. Commercial universal SPM [23] software, as well as specialized software (ImageJ, MountainsSPIP) widely use the methods (binarization, segmentation, contour analysis) implemented in the mentioned libraries. However, it is further shown that in some cases it is reasonable to use specialized methods for image processing. There are various special-purpose SPMs that can be used for nanoparticle size monitoring such as a multi-probe scanning microscope (an AFM with parallel active cantilevers) and large-sample SPM. The multi-probe scanning microscope can be used for fast investigation (several minutes) of a large area (several mm$^2$). However, it has a low resolution of about 40–50 nm [24], which makes it impossible to monitor nanoparticles with a diameter smaller than 40–50 nm.

A large-sample SPM can also be used for nanoparticle size monitoring. Such an SPM is equipped with advanced sample-positioning systems, which leads to an increase in the effectiveness of size monitoring. The use of a large-sample SPM is economically sound for monitoring nano-objects in mass-production items such as silicon wafers, catalysts, membranes, or filters. However, when the parameters of an item are changed, for example, a decrease in the sizes of a sample and nanoparticles, the large-sample SPM capabilities are the same or even worse than those of a conventional laboratory SPM. Another type of SPM providing possibilities for monitoring nanoparticle sizes is a universal highly automated SPM, for example, an SPM Next II [23]. Such an SPM allows selecting an area for investigation, automatic positioning of a probe to a selected area, scanning several overlapping areas and forming a panoramic image. However, in the automatic mode of operation, the software of commercial microscopes is not always capable of distinguishing and correctly determining the sizes of polydisperse powders, particles with over-lapping contours or considerable oscillations of the brightness function.

Based on the above-mentioned, we can conclude that the search for means to modernize existing SPMs (for example, open-access platforms such as Bruker Dimension Icon) for the automation of the size monitoring is of current importance especially when these means facilitate the development of SPM in general. This is confirmed by examples of a meaningful (considering the features of the objects under study) implementation of the tracking scan concept, implemented in a homebuilt SPM [25].

In the present study, we consider the formation of sampling on the basis of the processing of the image set of nanoparticles and their agglomerates deposited on a substrate for monitoring. These particular cases are characterized by the presence of the following limitations:

- Impossibility of the formation of the monolayer of particles guaranteeing their absolute distinguishability;
- Difficulty in searching for particles on a substrate.

The objective of the present work is the development of methods for particle-size monitoring in images acceptable in the conditions of the afore-described limitations and suitable for individual particles and the particles in agglomerates and multilayer coatings. The SPM is well-suited for studying nanoparticles deposited on a substrate from a gaseous phase or a liquid. The most difficult case for monitoring is the deposition of nanoparticles

from a liquid since it leads to the formation of multilayer, the relief irregularity of an obtained coating and the presence of agglomerates.

## 2. Materials and Methods

In the present study, there are examples of the SPM application for studying fullerene-containing powder (the particle sizes in the range of 0.7–100 nm) and nanoscale particles (the sizes in the range of 0.1–40 μm). The fullerene-containing powder was obtained in Udmurt Federal Research Centre (Izhevsk, Russia) by electrochemical deposition and dispersion at high pressure. The nanoscale particles were obtained in the multi-stage centrifugal mill designed and assembled in Udmurt Federal Research Centre. Highly oriented pyrolytic graphite (HOPG) and polished ceramic plates (1 × 1 cm) were used as substrates. The substrates were purchased from the NT-MDT company.

The electrochemical deposition of fullerene-containing powders on HOPG was performed from a fullerene solution (the concentration ~150 μg/g) in paraxylene with sodium metal at the voltage of 100–150 V and galvanic current of 10–30 μA. HOPG was preliminarily subjected to mechanical cleaning using scotch tape.The cooper particles dissolved in polystyrene were deposited on the plate surface by the spin-coating method. The plate was preliminarily cleaned with ethanol.

In order to automate the investigations, a microscope based on a commercial P47 SPM was used. The field of view of the microscope was 100 × 100 × 6 μm (±10%), and the minimum scanning step was 0.006 nm. It was upgraded as follows. Firstly, the manual positioning screw drivers were motorized. Secondly, the software designed for motion control and image processing was upgraded. To minimize any actions upon a surface the modes of tunnelling microscopy and semi-contact atomic force microscopy were used.

## 3. Size-Monitoring Method with the Use of the SPM

The nanoparticle-size monitoring with the use of the SPM can be described as follows. Firstly, a surface area is selected (for example, with the help of a video camera); after that, the selected area is scanned several times at high resolution. Let us consider the case when the size of the studied area exceeds the microscope field; it is rather common considering that the resolution of an optical microscope is 1 μm and the size of high-resolution scanners is 1–3 μm. Traditionally, the scanning of surface areas exceeding the field is performed using overlapped scans and a sample displacement. Before the displacement, the probe is taken away from the surface and brought back after the displacement. The scanning process is performed until there are a required number of particles on the acquired images, the sizes of which form the statistics for monitoring. The scans can be uniformly arranged over the specified area with a fixed overlap (uniform scanning, Figure 1a) or along the line with an increased concentration of controlled particles (tracking scan, Figure 1b). For example, when particles are deposited on the HOPG surface, the most probable site of their allocation is the boundaries of terraces. In this case, the tracking scan provides a considerable decrease in the monitoring time. At such scanning, the displacement of scans is performed by a scanner or motorized sample platform. Here it is necessary to consider that the error in the positioning by a motorized platform is significantly bigger than that in the case with a scanner. The acquired scans (images) are stitched into a single image before processing or are processed separately while the repeating results in the overlap regions are excluded.

The uniform scanning of a studied area is sufficiently elaborated by the microscope and software manufacturers. In the present study, we turn our attention to tracking scans. Tracking scanning allows to significantly save the time of scanning. The amount of savings provided in [25] amounted to up to 67% of the scanning time [25]. For example, in the case shown in Figure 1, the time savings is 5/9 or 55%. It should be noted that significant time savings are also achieved by automating the pre-processing of images and the selection of the trajectory of the displacement of the field of view, carried out by the experimenter manually.

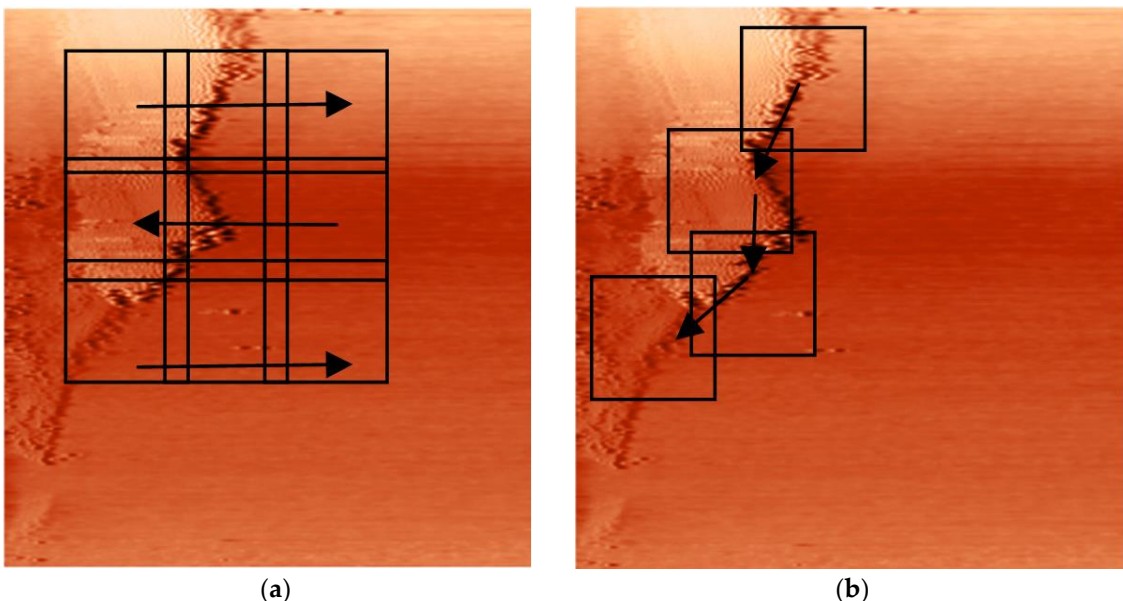

(**a**)                           (**b**)

**Figure 1.** Options of the SPM operation in the process of the size monitoring: (**a**) uniform scanning; (**b**) tracking scanning.

Our approach to particle size monitoring, founded on tracking scan, is based on solving the following tasks:

(1) Particle detection: image segmentation, particle allocation, the determination of the particle sizes and coordinates;

(2) Computation of the sample displacement value providing the coordinate coupling of two overlapping images and the continuation of the particle-size data acquisition;

(3) Computation of the "global" coordinates of the particle centres for excluding repeating measurements in the areas of the image overlapping.

To solve the first task, the modern SPM software contains the tools for particles detection and determination of their sizes, or such tools are in specialized software. For example, the NT-MDT microscopes (the software ImageAnalysis) contain a GrainAnalysis function, and in the specialized software MountainsSPIP there is a Grain/Pore Analysis function. The drawback of the above tools is the necessity of manual adjustment. In addition, these tools are based on the standard tools for image binarization and segmentation (Watershed [26], Otsu's method [27]), which decreases the accuracy of the size determination for superimposing particles and particles with different degrees of brightness. The difference in brightness is possible even when particles are deposited from a liquid solution on atomically smooth substrates. This is due to nano- and microparticles that tend to form polydisperse agglomerates and a surface with an irregular relief. There are more advanced methods for segmentation such as the parametric (with the estimation of the parameters of the grey-level distribution of each class of objects in an image [28,29]), variational [30], neuronet [31] and correlation methods [32]. All the above methods are limited in application and they are not always available in open-ended libraries such as OpenCV. In addition, the parametric and variational methods are distinguished by their considerable computational complexity and complexity of implementation; regarding the neuronet methods, sometimes it is difficult to provide a sufficient amount of the training material. The investigations of the segmentation methods with the use of open freeware (ImageAnalysis, ImageJ, WSxM) show the following: The images segmentation using conventional methods based on the intensity function is rather difficult for complex images with particle agglomerate. The difficulty of segmentation leads to the impossibility of accurately determining the size and coordinates of particles and the implementation of the proposed monitoring method. In

the present study, we consider the approach of image segmentation based on the surface curvature function.

For solving the second task, motorized sample platforms are used; as a rule, microscopes for studying large-size samples are equipped with such platforms. In this case, the larger the platform travel range is, the larger the minimal step of positioning is (Table 1). Judging by the value of the minimal step, the sample-positioning platforms in a commercial SPM are designed for use with an optical microscope (resolution of about 1 μm). The use of such systems with a scanner having a travel range of less than 1 μm is difficult.

**Table 1.** Characteristics of probe microscopes.

| SPM Model | Company | Range, mm | Minimal Step (Resolution), μm |
|---|---|---|---|
| C3M Next II [23] | NT MDT | $5 \times 5$ | 0.3 |
| AFM5500M [33] | Hitachi | $100 \times 100$ | 2 |
| Dimension Icon [34] | Brucker | $180 \times 150$ | 2 |

The presence of an error in the positioning of the motorized platform causes an uncertainty in the value of the scan overlapping, which makes difficult the solution of the third task, the determination of the probe and particles coordinates in a single global coordinate system. To solve the task, in an SPM the sensor values or the tools for the overlapping-image processing can be used. When sensors are used, each image and each particle have their unique coordinates, which allows for excluding repeating results in the overlap region. In this case, the main problems are as follows:

- Sensors are not always present in the SPM positioning system, and sometimes their integration in this system is constructively impossible and impractical;
- The sensor accuracy may be insufficient for the unambiguous establishment of the correspondence of the centres of the particles.

Software (algorithm) for stitching or coordinate coupling of overlapping images can be an alternative to sensors. When algorithmic methods are used, the accuracy of the coordinate determination depends on the resolution of the scanner and image and not on the precision of the platform or sensor. When the images are stitched, a single image is formed, and each particle is unique within the image. At the coordinate coupling, the mutual shift (displacement) and rotation of the images are determined. Using these values, the local coordinates of particles in each image can be combined into a common coordinate system. The cases are known of the combined use of the sensor values and the images for high-precision probe positioning. For example, in the SPM NEXT II [23], the sample positioning by the motorized platform is additionally controlled by a high-resolution video camera.

Detectors and descriptors of key points are widely used for matching and stitching of images. Binary descriptors (BRIEF, ORB and BRISK) refer to low-level ones and they are sensitive to distortions associated with the change in scale, shape, and point of view [35] which are quite probable in an SPM. The SPMimages can contain distortions even after processing, which influence the result of the measurement of object sizes. For example, an image can contain lines significantly different from neighbouring lines due to the probe-surface interaction or l/f noises [36]. There can also be brightness oscillations due to the final speed of the regulating system, surface contamination, vibroacoustic disturbances, and imperfect scanning system. Such disturbances influence the object shape and the position of key points. More complex, transformation invariant descriptors such as SIFT [37], SURF [38] can also be ineffective in the conditions of disturbances and distortions characteristic of the SPM.

The analysis shows that specialized methods should be used for image matching instead of conventional ones. For example, it is logical if the centres of key points coincide with the particle centres and the information on the particle size is included in the key

point descriptor. It is also advisable to use the values of the platform sensors for determining the mutual orientation (displacement, angle of rotation) of the images (for example, in the topoStitch program [39], prior information on the image values can be used for image stitching).

Considering the difficulties in solving the problem of particle agglomerate size monitoring by conventional methods, we propose a new integrated approach based on the use of:

- The surface curvature detectors for correct image segmentation and particle detection;
- The detection results for overlapping images coordinate matching and sample displacement computation;
- The complexing of displacement sensors and the determined images overlap value for refinement of results.

## 4. Detection of Particles Using the Surface Curvature Function

For particle detection, the curvature function of surface or plane curves can be used. However, conventional methods for the curvature computation use derivatives; thus, the field of their application is limited to smooth surfaces [40]. Therefore, the development of new methods for the computation of the curvature of the SPMimages considering their peculiar features is of high practical value. Let us consider one of the possible realizations of the method for computing the surface curvature [41,42] and particles sizes in the SPMimages.

To determine the curvature the neighbourhood of each point in each line of the SPMimage $Z(x)$ is approximated by the segment of the adjacent circle. The segment height $\varepsilon = AB$ is a constant value for the entire image, and the chord length $H = P_1P_2$ is a variable (Figure 2).

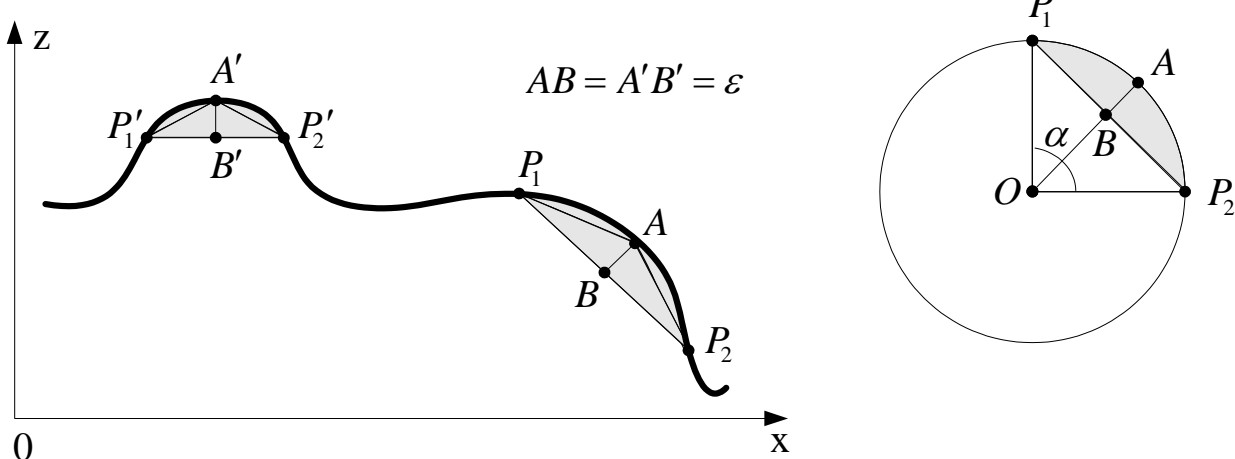

**Figure 2.** Schematic view of the curvature detector operation.

The radius $R$ of the particle osculating circle is computed from the figure $\Delta P_1OB$:

$$\left(R - \varepsilon^2\right) + \frac{H^2}{4} = R^2, \ R = \frac{H^2 + 4\varepsilon^2}{8\varepsilon} \tag{1}$$

The algorithm of the curvature computation is as follows. The points $P_1$ and $P_2$ are incrementally moved away from A with the step of 1 pixel along the $x$ axis until $AB < \varepsilon$. After each step, the coordinates of points $P_1$ and $P_2$ are determined and the general equation

of the line passing through these points is computed. The value $AB$ is determined by the following formula:

$$AB = \left| \frac{ax + by + c}{\sqrt{a^2 + b^2}} \right|; \ AB = \left| \frac{c}{\sqrt{a^2 + b^2}} \right|_{(x=0, y=0)} \tag{2}$$

where $a$, $b$, and $c$ are the coefficients of the general equation of the straight line drawn through $P_1$ and $P_2$. If the inequality $AB < \varepsilon$ ceases to hold, then the chord length is calculated. The $H$ values for the image lines and columns form the functions $H(x)$ and $H(y)$.

For the images with a low level of distortions, the centre of a particle is the point of the coincidence of the local extrema $H(x)$ and $H(y)$. The particle radius is computed by Formula (1). In the presence of distortions, the particle centres can be determined by the local extrema of the function:

$$W^+(x, y) = \begin{cases} H_y(x) + H_x(y), & (H_y(x) \geq 0) \wedge (H_x(y) \geq 0), \\ C, & \text{otherwise,} \end{cases} \tag{3}$$

where $C \geq \max (H_y(x) + H_x(y))$ is a constant. In the presence of several close-spaced extrema $W^+(x, y)$, the dominant extremum is chosen. The curvature detector is sensitive to image noise. Therefore, preliminary smoothing is required.

## 5. Determination of the Parameters of the Mutual Orientation of Overlapping Images

The existing methods of image stitching can require considerable (up to 50% [43]) overlapping of images which is necessary for the allocation of the sufficient number of common key points in the images $Z_0$ and Z. When particle sizes are determined with the use of several overlapping images, the result of the stitching process is not important. It is important that the parameters of the mutual orientation of overlapping images, displacement and rotation, are determined; this permits the determination of the overlap region and exclude the repeating results of the measurements relating to the region. The determination of the parameters is more exact when there is a sufficient number of common points. For the homography, the number of such points should not be less than 4 and at the use of the method of image matching by key points' pairs (IMBKPP) [44], it should be 3. Let us consider the realization of the IMBKPP method where the detected particle centres are taken as key points.

In order to compute the displacement and rotation of the images, we used the key points' pairs of the images, which are made up of the key points from the total set of key points. Each key points pair (line segment) is described by the coordinates of the ends $(x_1, y_1)$ and $(x_2, y_2)$ and the length $l$. For the computational shortcut, the displacement values $\Delta_x$, $\Delta_y$ and the rotation angle $\varphi$ are computed only for the pairs of points with the same length. The order of the computations is provided in [44]. The $\Delta_x$ and $\Delta_y$ values form a two-dimensional array $M(\Delta_x, \Delta_y)$ where the number of certain displacement values $(\Delta_x, \Delta_y)$ is entered into the cell with the coordinate $(\Delta_x, \Delta_y)$. The set of the values of the rotation angles forms a histogram of iteration $N(\varphi)$ for the range from $-180°$ to $180°$. The coordinates of the maxima in $M(\Delta_x, \Delta_y)$ and $N(\varphi)$ determine the displacement and rotation of the images.

During the size monitoring process, to minimize losses of time for scanning the same surface areas, it is practical to restrict the overlap region of the sequentially acquired images so that there would not be more than 3–5 particles. However, it can lead to a decrease in the overlapping coefficient by 10–15%. As the practice showed, the background elements (disturbances) cause the appearance of false key points and several local maxima in the array $M(\Delta_x, \Delta_y)$ which makes the computation of the mutual orientation parameters of two images difficult.

In the simplest case, a key points pair is described with the use of the descriptor D1 which represents the coordinates of the ends $(x_1, y_1)$ and $(x_2, y_2)$ and the length $l$ of the line segment connecting them: D1 = {$(x_1, y_1)$, $(x_2, y_2)$, $l$}. To improve the accuracy of the pairs'

correspondence, the more informative pair descriptor is recommended: D2 = {($x_1$, $y_1$), ($x_2$, $y_2$), $R_1$, $R_2$, $l$}, where $R_1$ and $R_2$ are the radii of the particles, the centres of which coincide with key points. In this case, the correspondence search algorithm checks both the length equality of the pairs in the images and the validity of the condition:

$$
(\left|R_1 - R_i + R_2 - R_j\right| < 2t) and (((\left|R_1 - R_i\right| < t) and \\
and(\left|R_2 - R_j\right| < t)) or ((\left|R_1 - R_j\right| < t) and(\left|R_2 - R_i\right| < t)))
$$
(4)

where $R_1$ and $R_2$ are the radii of the curvature of the key points pair of the first image; $R_i$ and $R_j$ are the radii of the curvature of the key points pair of the second image; $t$ is the tolerance value. It is also possible to exclude the key point pairs with a nonzero rotation angle from the correspondence analysis since the second image was acquired after the sample displacement in two orthogonal directions. The pair descriptor in this case has the form D3{($x_1$, $y_1$), ($x_2$, $y_2$), $R_1$, $R_2$, $l$, $\varphi$}. Descriptors D1–D3 are sensitive to the scale of the image. Their use implies the maintenance of the field of view and the resolution of the microscope during the tracking scan.

## 6. Complexing of the Methods of Images Overlap Estimation

As was mentioned earlier [44], the IMBKPP method provides good results for high-quality images with a high overlapping coefficient and a considerable number of particles in the overlap region. When the overlapping coefficient is low, the number of common key points of the images decrease; the maximum value of the array M($\Delta_x$, $\Delta_y$) may be small (only 1–3units), and the number of the equivalent maxima exceeds 1.

To avoid this, when choosing the offset of the subsequent scan, it is necessary to ensure that there are more than three particles in the overlap area. For example, we usually set the overlap region dimensions so that there were no fewer than 10 particles. The reason for this is that when the density of the similar-size particles is high, the probability of a false correspondence of key point pairs (especially short ones) increases. The technique for determining the overlap region is as follows: The overlap of 35% is set along X and Y axes (the area of the overlap region is 12.5%). If in the overlap region the number $N_{OV}$ of particles, allocated in the overlap region, is smaller than required, the overlap region is gradually increased along one coordinate until the required value of $N_{OV}$ is attained. When necessary, the overlap region can be increased along the other coordinate as well. Contaminations or image noise have an effect on the number of particles. If there is a chance of such contamination, then it is necessary to increase the overlap area.

Thus, to increase the reliability of the determination of the scan overlap parameters, several methods for estimating these parameters should be used [45]: first, algorithmic methods for the image processing; and second, the values of the sensors mounted on the sample-positioning platform or displacement values obtained on the basis of the signals controlling the sample-positioning platform drives. In our SPM we use the positioning platform where the leadscrews for manual control are equipped with inertia piezo-drives and sensors [45].

If there are several maxima in the array M($\Delta_x$, $\Delta_y$), the sensor values limit the valid region of the M($\Delta_x$, $\Delta_y$) and permit to select the truer extremum. In this case, the accuracy of the final result computed by the software is considerably higher than the sensor accuracy. The sensor resolution should correspond to the value of the linear scans overlapping along one of the axes and make up 10–15% of the scan size. The accuracy of the software method is 2–3 pixels, which is 0.8–1.1% for the scan of 256 × 256 in size.

## 7. Result and Discussion

We compare the proposed method of particle size monitoring with a GrainAnalysis function of a commercial P47 microscope. We use three pairs of agglomerate images: a test pair in Figure 3a,b; images of agglomerates on a flat surface in Figure 4a,b; images of agglomerates on a surface with a complex relief Figure 5a,b. Each test images contain 47 particles. The particles size on the images is 10 pixels. The images differ in the particle

brightness. The test images were modelled, the images in Figures 4 and 5 were received in semi-contact mode (scanning time ~4 min).

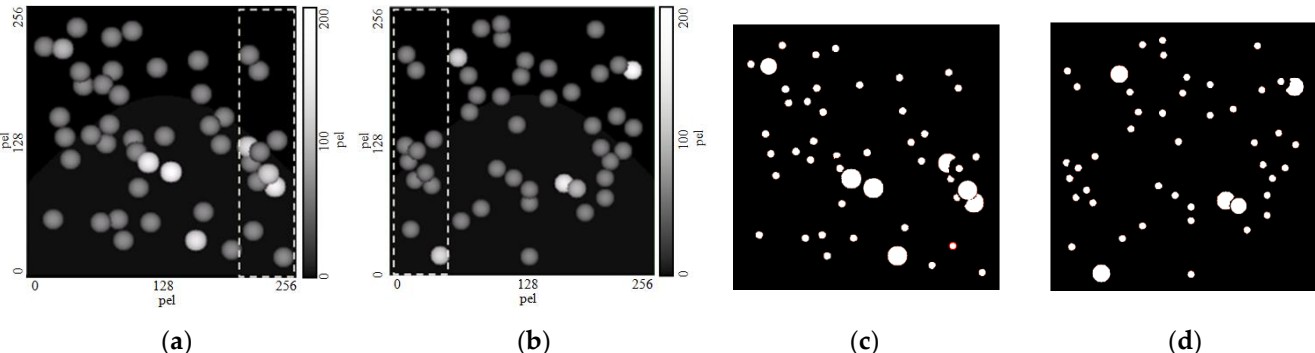

**Figure 3.** Overlapping test images (256 × 256 pixels, Z range—200, *x*-axis offset 190 pixels): (**a**,**b**) source images with marked overlap area; (**c**,**d**) the results of segmentation by GrainAnalysis.

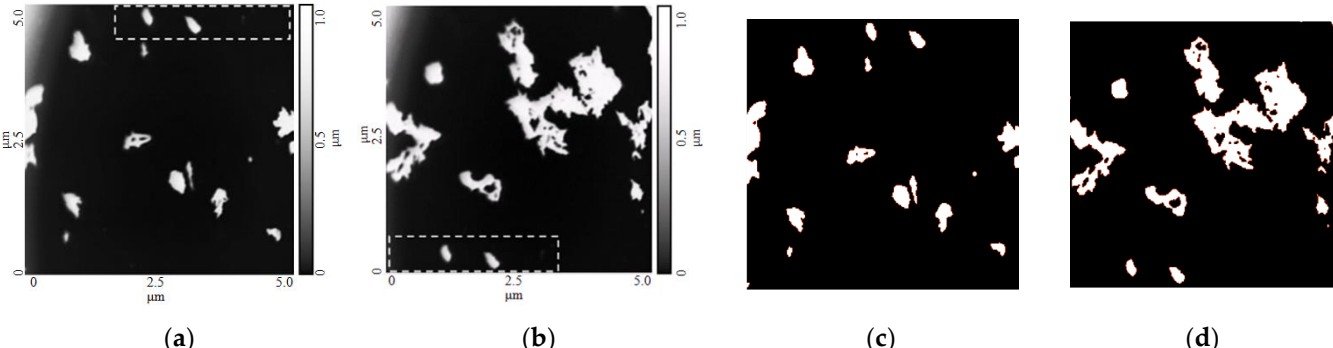

**Figure 4.** Overlapping copper microparticles images (256 × 256 pixels): (**a**,**b**) source images with marked overlap area; (**c**,**d**) the results of segmentation by GrainAnalysis.

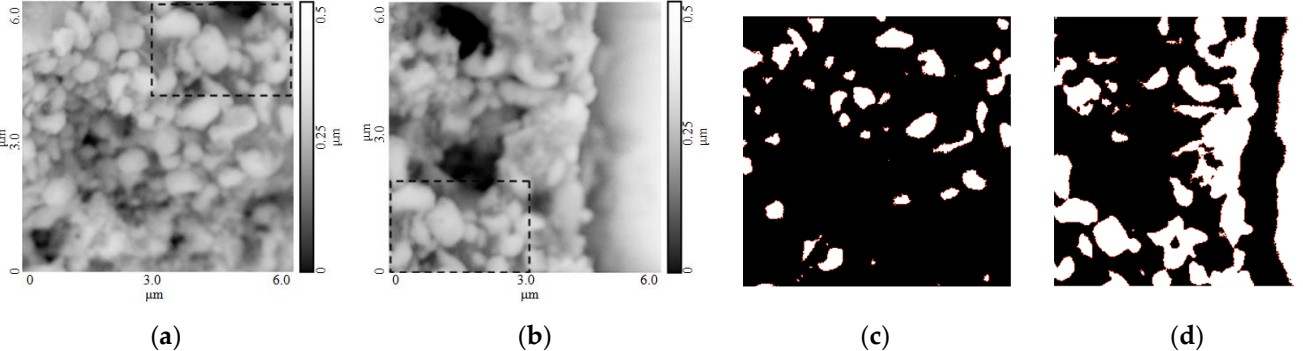

**Figure 5.** Overlapping fullerene' images (256 × 256 pixels): (**a**,**b**) source images with marked overlap area; (**c**,**d**) the results of segmentation by GrainAnalysis.

The results of segmentation of test images obtained using the GrainAnalysis tool are shown in Figure 3c,d. For quantitative estimates, we used the number Nd of detected particles and their average size Rd, as well as the average size relative deviation $\varepsilon$. It follows from the comparison results (Table 2) that the conventional method did not allow the correct detection of agglomerate particles. At the same time, the separation of agglomerates into particles may be impossible or may be accompanied by a decrease in size. This is mainly due to the limited capabilities of conventional methods. The comparison of the segmentation results obtained by the curvature detector and conventional methods (Figure 6) shows

the effectiveness of the curvature detector for the segmentation of agglomerate images. It detected all 47 particles in the test image, unlike the conventional methods. In addition, the average relative deviation of the particle sizes (in the test images) was no more than 3% at the same time. Unfortunately, it is difficult to assess the accuracy of the detector on more complex images (Figures 4 and 5a,b). The reason for this is the lack of prior reliable information about the sizes of particles and agglomerates.

**Table 2.** Comparison of segmentation results by the curvature Detector and the GrainAnalysis tool (the results for the Grain Analysis tool are given for Figure 3c,d in parentheses).

| Parameter | Curvature Detector | Grain Analysis Tool |
|---|---|---|
| Nd | 47 | 47(47) |
| Rd | 9.84 | 8(9) |
| $\varepsilon$ | 0.016 | 0.2(0.1) |

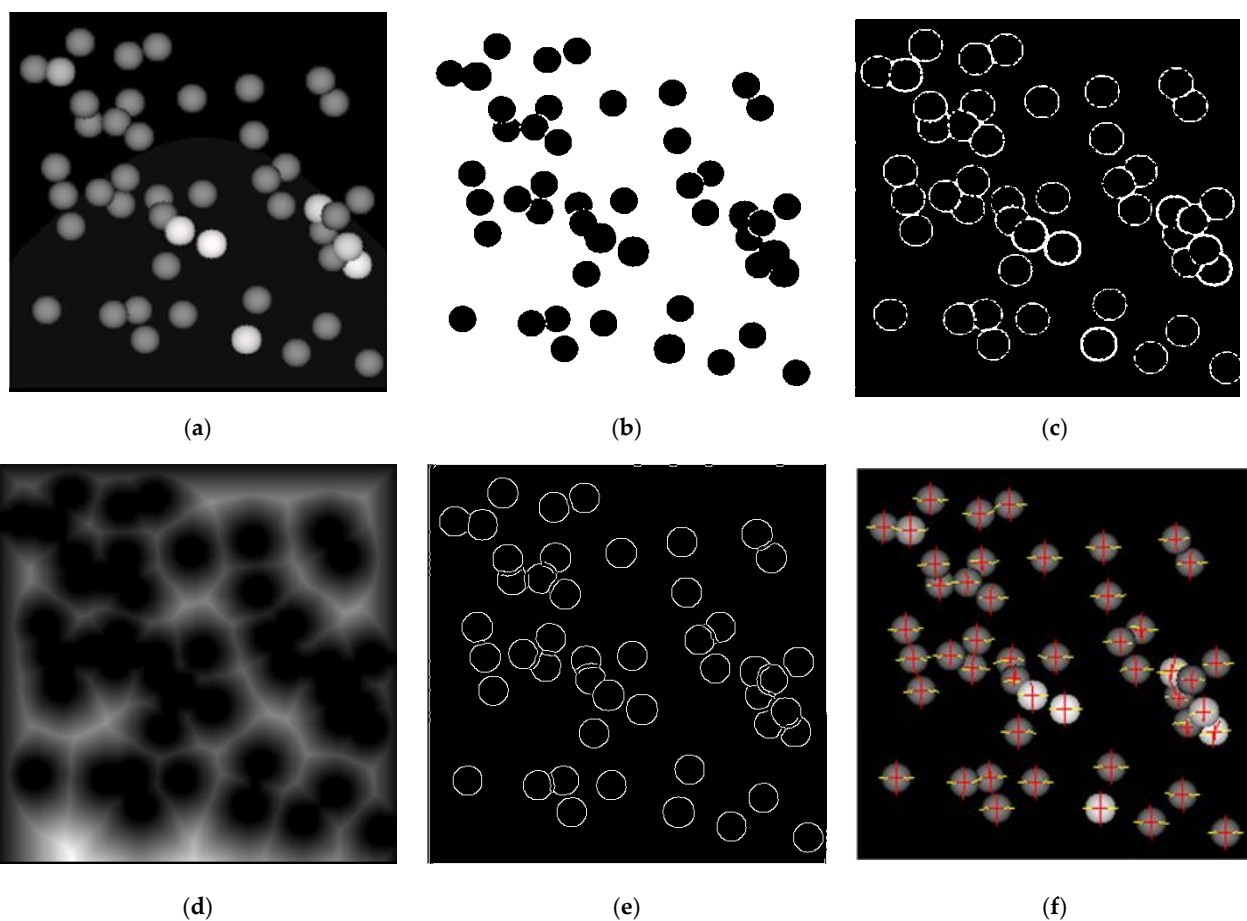

(**a**)  (**b**)  (**c**)

(**d**)  (**e**)  (**f**)

**Figure 6.** Comparative results of the segmentation on test image: (**a**) the initialimage (the scale according to Figure 3a), containing 47 particles with a radius of 10 pixels; (**b**) segmentation by Otsu's method; (**c**) segmentation with the use of the adaptive threshold technique; (**d**) segmentation by Watershed method; (**e**) segmentation by Canny method; (**f**) the centres of the particles, and the zones of maximum curvature, obtained using a curvature detector.

To compare the methods of particles monitoring on the complex images (Figures 4 and 5), we use threshold segmentation performed using the Grain Analysis tool. The threshold was adjusted manually to the value at which the agglomerate particles in the overlap area are separated from each other. A comparison of the segmentation results with the GrainAnalysis tool (Figures 4 and 5c,d) and the curvature detector (Figures 7–9) shows

that the detector highlights more key points in the overlap zone, which makes it easier to overlap images in coordinate coupling. Figures 7, 8 and 9a,b shows two overlapping images and the computed results for the displacement and rotation. The scale of the images (Figures 7–9) corresponds to Figures 3–5. The array M is displayed in the form of the two-dimensional map of the displacement values (Figures 7, 8 and 9c: dot designates a non-zero element of the array M, and cross designates a maximum) and the array $N(\varphi)$ is shown in the form of a diagram (Figures 7, 8 and 9d). The orientation parameters $\Delta_x$, $\Delta_y$ and $\varphi$ are determined from the positions of the maxima of the M and N arrays.

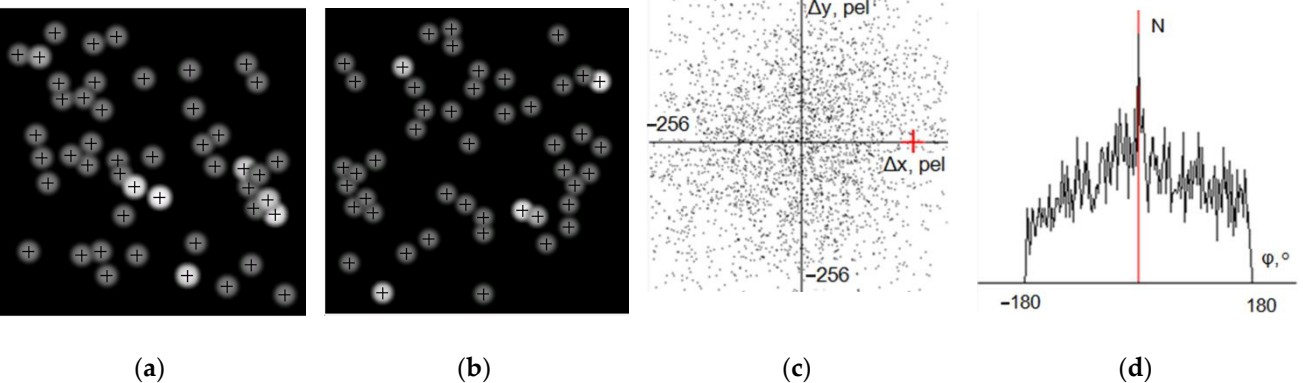

(**a**)                     (**b**)                     (**c**)                     (**d**)

**Figure 7.** Results obtained by the IMBKPP method for the test images: (**a**,**b**) source images with marked particle centres; (**c**) the map of displacement values; (**d**) the histogram of rotation angles.

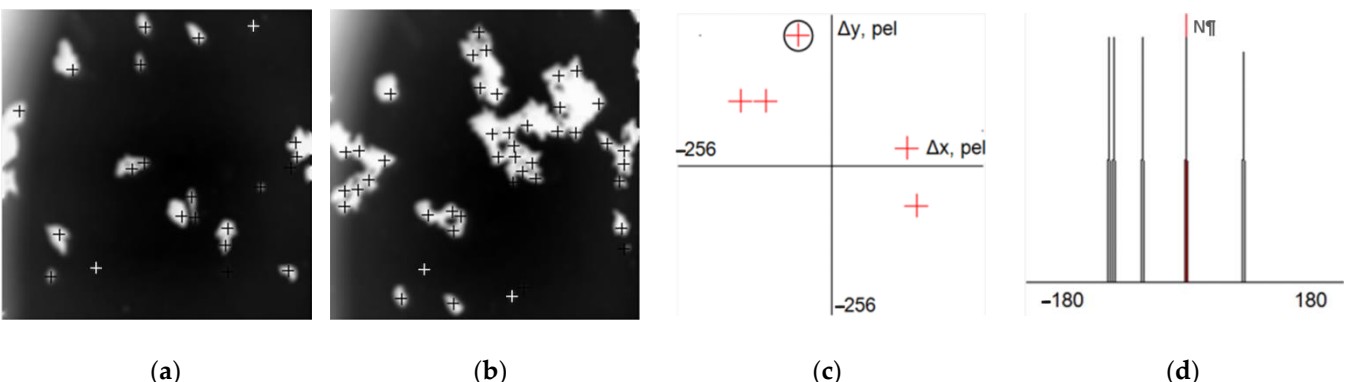

(**a**)                     (**b**)                     (**c**)                     (**d**)

**Figure 8.** Results obtained by the IMBKPP method for the copper microparticles' images: (**a**,**b**) source images with marked particle centres; (**c**) the map of displacement values; (**d**) the histogram of rotation angles.

Figures 7–9 present the curvature detector and IMBKPP which achieved satisfactory results (the specified *x*-axis offset is correctly determined) on images, on which the conventional methods (such as SURF descriptor and FLANN library) showed unsatisfactory results. The overlapping areas of the images (Figures 7–9) correspond to Figures 3–5. Figure 8 confirmed the operability of the proposed monitoring approach for images with low overlap, which required the use of a sensor to determine the true extreme of the displacement (Figure 8c, encircled). Figure 9 confirmed the operability of the proposed approach in images with complex relief. The detector's operability was determined by visual comparison of similar key points of overlapping images, as well as an assessment of their position on the image and histogram.

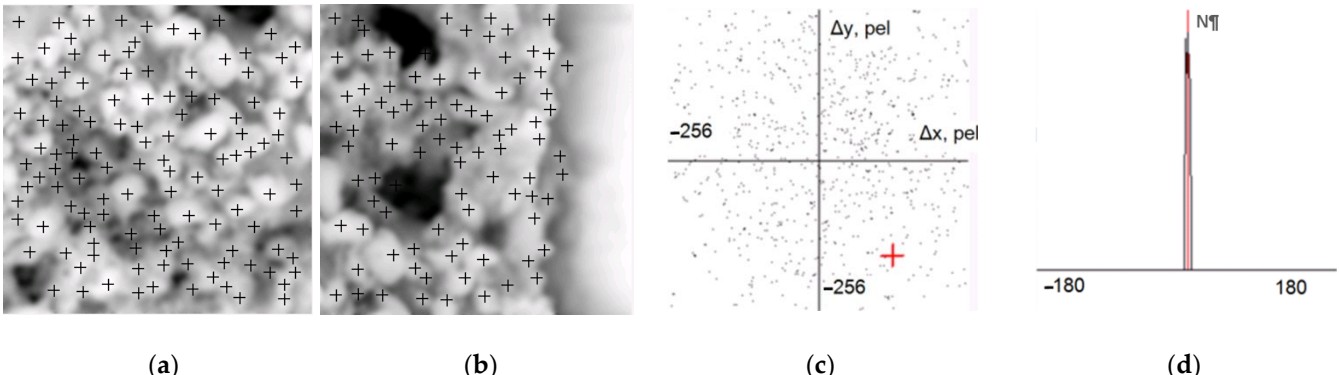

**(a)**      **(b)**      **(c)**      **(d)**

**Figure 9.** Results obtained by the IMBKPP method for fullerene-containing powder images: (**a**,**b**) source images with marked particle centres; (**c**) the map of displacement values; (**d**) the rotation angle histogram.

Figure 8 shows that the centres of several densely packed agglomerates, determined by the curvature detector, may not match the visually detected centres. This problem can be solved by increasing the image resolution or reducing the microscope field of view. The number of detected particles increases in proportion to their concentration if the image resolution is sufficient. Images of polystyrene spheres coated by gold (Figure 10), obtained from the scan gallery [46], and Table 3 confirm this fact. The size of the image elements that can be detected should not be less than 4–5 pixels. This is illustrated by Figure 11 [46], where some small elements, located at a short distance (4–5) pixels from other elements, were not detected.

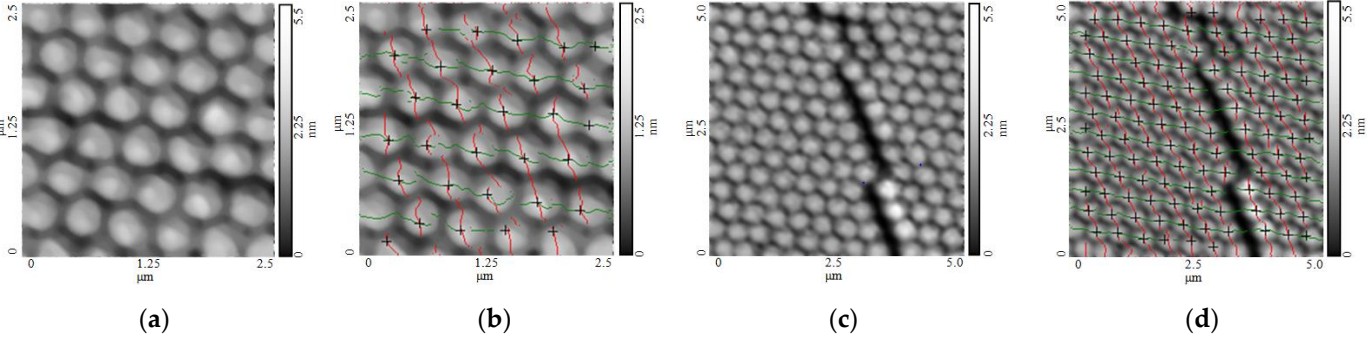

**(a)**      **(b)**      **(c)**      **(d)**

**Figure 10.** Images of polystyrene spheres coated by gold: (**a**) source images 2.5 × 2.5 µm; (**b**) images with marked centres; (**c**) source images 5 × 5 µm; (**d**) images with marked centres.

**Table 3.** Detection results on images with different particle concentrations.

| Parameter | Figure 10a | Figure 10c |
|:---:|:---:|:---:|
| Nd | 30 | 118 |
| Rd | 17.98 | 8.98 |

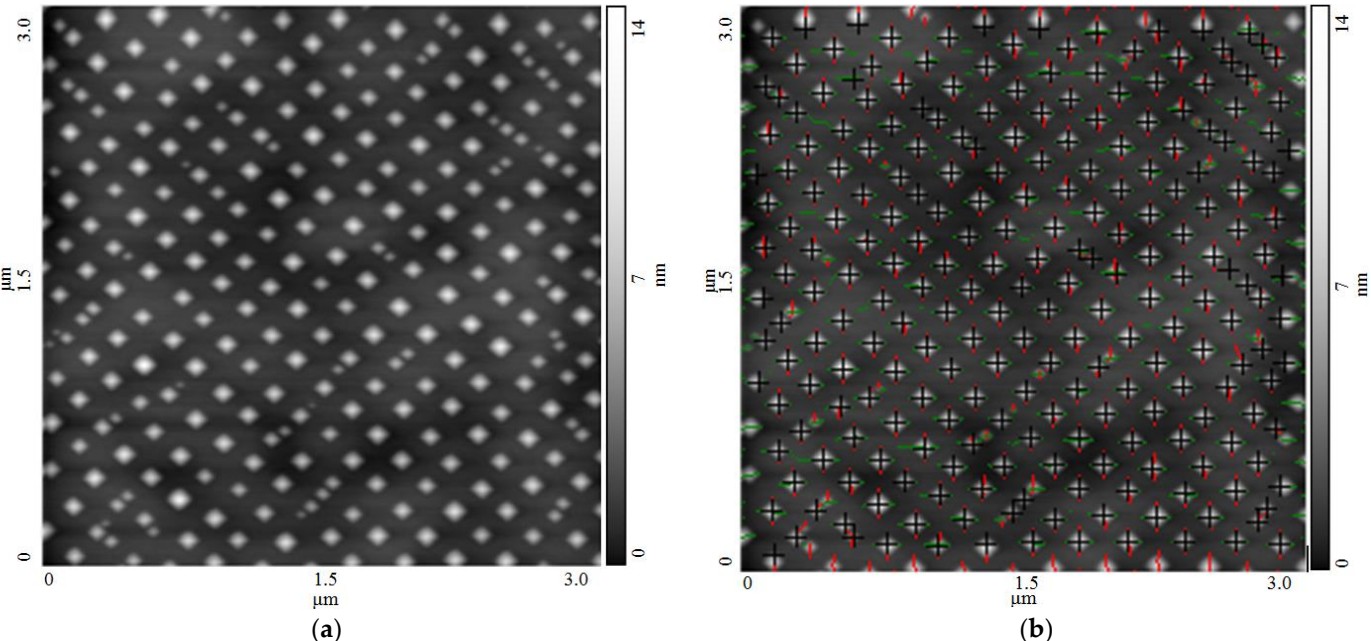

**Figure 11.** Array of ordered Ge(Si) pyramid islands, grown on relaxed SiGe buffer layer [46]: (**a**) source images 3 × 3 μm; (**b**) images with marked centres.

## 8. Conclusions

In order to automate the monitoring of agglomerate sizes, an approach to solving the task of surface tracking scan in probe microscopy is formulated. A feature of the proposed approach is the use of information about the particlesize and coordinates to determine the subsequentscan positions and generate statistics of the results.

The use of a surface curvature detector improved the accuracy of determining the coordinates and sizes of agglomerated particles. In combination with the use of motion sensors, this increases the reliability of stitching tracking images and allows to generate correct size statistics.

The proposed approach was tested on polystyrene particles, Ge(Si) islands, fullerene-containing powders, and agglomerates containing nanoscale diamonds and copper particles. The possibility of the separation of individual particles and the particles in the agglomerate composition was confirmed.

The results presented in the study can be used in other types of microscopy and inspection methods with the use of images.

An example of the implementation of algorithms in the Pascal (Delphi) is available at the link http://iam.udman.ru/ru/node/6653 (accessed 9 January 2022).

**Author Contributions:** Conceptualization, P.G. and A.K.; methodology, E.S. and T.K.; software, P.G. and E.S.; validation, P.G. and E.S.; formal analysis, P.G. and T.K.; resources, T.K.; data curation, P.G. and A.K.; writing—original draft preparation, P.G.; writing—review and editing, A.K. and T.K.; visualization, P.G. and E.S.; project administration, T.K. and A.K.; funding acquisition, T.K. All authors have read and agreed to the published version of the manuscript.

**Funding:** This work was supported by the Slovak Ministry of Education within project VEGA No. 1/0823/21 and by the Slovak Research and Development Agency under contract No. APVV-18-0316.

**Institutional Review Board Statement:** Not applicable.

**Informed Consent Statement:** Informed consent was obtained from all subjects involved in the study.

**Data Availability Statement:** In the paper the publicly archived gallery of scans [46] was used.

**Conflicts of Interest:** The authors declare no conflict of interest. The funders had no role in the design of the study; in the collection, analyses, or interpretation of data; in the writing of the manuscript, or in the decision to publish the results.

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
