# Peer review of "Particle and Particle Agglomerate Size Monitoring by Scanning Probe Microscope"

_applsci, doi:10.3390/app12042183_

Round 1
Reviewer 1 Report
The paper addresses the characterization of nanoparticles as captured on images (their sizes, number, positions) with respect to present agglomerates.
The topic is of great interest to the readership and fits well within the scope of the journal. It is well written however, there is a place for the improvement of the scientific soundness.
Please address the following:
Figure 6 f looks like a horizontal flip! Please check, that is the important result, the essence of this paper
Figure 8: should there be an overlap between a and b visible, what is the the notion of the overlap area, I don't see the marks. Please make it more clear
Author Response
Dear Reviewer,
thank you so much for your valuable comments aimed at improving the quality of our article. For more details, please see the attachment.
Best regards,
Authors

Reviewer 2 Report
The reviewed manuscript entitled ‘Particle and particle agglomerate size monitoring by scanning probe microscope’ by Gulyaev et al. is related to the new monitoring method of a size of different nanoparticles on the solid substrates using scanning probe microscopy. The results presented in the manuscript can be useful to the wide group of researchers investigating nanoparticles in thin films. However, the manuscript should be better organized and improved in terms of the text edition. Below the advices that may help to improve the manuscript are listed.
- Introduction section should provide wider description of the state of the art. Some information from Size-monitoring method with the use of the SPM section should be here transferred. Moreover, one or two sentences cannot create the separate paragraph.
- The most of the information from the first paragraph of Materials and Methods section should be transferred to Introduction
- In Materials and Methods section, there is a lack of some important information needed to repeat described experiments such as producer of used substrates, their cleaning procedures, synthesis procedure or purchase place of the nanoparticles used during research
- In the SPM images a scale bar should be added.
- Captions of the tables 2 and 3 should be more accurate and exact.
- Links to website should not be added in the main text.
- References to figure in the main text should be unified.
Author Response
Dear Reviewer,
thank you so much for your review and remarks that helped us improve our article. For more details, please see the attachment.
Best regards,
Authors

Reviewer 3 Report
This work presents an algorithm to minimize image allocation overlapping for speed size measurements. Core claims are well supported by the experiments and I think the current scope well fit the journal's area.
Author Response
Dear Reviewer, thank you so much for your review.
Best regards, Authors
This manuscript is a resubmission of an earlier submission. The following is a list of the peer review reports and author responses from that submission.
Round 1
Reviewer 1 Report
This manuscript deal with the determination of the size of nanoparticles with AFM.
First it describes the problems related with the available setups and softwares. Then it proposes a method to solve the issue.
Finally it applies the method to a single case of a surface with nanoparticles dispersed but often bundled to form a rather rough surface.
I find the problem rather interesting and relevant to current research.
It describes well the problems connected with the commercial systems and of the softwares now available.
The method proposed is also interesting, though it does not discuss the issue of tip convolution with the real surface, that is the more relevant the more the local curvature is high.
The example proposed is not conclusive in my opinion. The surface is rather rough and as mentioned above the method proposed may be not applicable. Most of all, it cannot be cross checked with other methods.
I would suggest to carry out some more analysis on more meaningful samples, starting from simple ones such as dispersed nanoparticles on a flat surface in order to finish with rougher surfaces with aggregates.
Other issues:
The manuscript should be strongly revised. The english is poor and this is more relevant where the description needs to be more accurate.
In the captions, the size of the area and z-scale are not reported in the standard notation. Please indicate clearly and separately the scan area and the z-scale.
Additionally the number of sampling points is often not clearly reported and thus the sampling distance, i.e. the distance value between two adjacent points. This is quite relevant as it limitates the minimum resolution achievable.
Author Response
Dear Reviewer,
we are very grateful for your commentaries aimed at improving the article.
We tried to respond to them as best we could and added recommended improvements and explanations into the manuscript. For further details, please see the attachment.
Best regards, Authors

Reviewer 2 Report
The manuscript by Gulyaev et al is written in an incomprehensible way. Therefore, it is not possible to review it in a proper, fair manner:
-The introduction, where the motivation of the work is stated, is confusing, not at all clear.
-I cannot distinguish clearly enough which are the novel contributions/setup of the SPM, particle analysis method developed by the authors and which parts correspond to existing methods integrated in commercial SPMs and softwares
-Is it the SPM the authors speak about, commercial or homebuilt? If it is homebuilt, are there parts/components that belong to commercial companies? Please, develop this part in the manuscript
-"Tracking scanning mode" is developed for the first time by the authors or is it available commercially, implemented in other commercial SPM images analysis softwares, following the same or similar processing procedure?
-It is not clear enough what are the strong points of the proposed analysis method by the authors in comparison with others because it is difficult to separate the authors contributions with the explanation given about other existing techniques. Sometimes, the manuscript looks like a "review" of existing methods, rather than an original article. Which are the types/the range of samples/materials that can be analysed using the methods proposed by the authors?
For all the above mentioned reasons, I strongly, yet kindly suggest that the authors re-submit the manuscript again after substantial amendment, taking into account the points developed, questioned in the present evaluation, to make the present work potentially considered to be published by the journal "Applied Sciences".
Author Response
Dear Reviewer,
we are very grateful for your commentaries and questions aimed at improving the article.
We tried to respond to them as best we could and added recommended improvements and explanations into the manuscript. For further details, please see the attachment.
Best regards,
Authors

Round 2
Reviewer 1 Report
Some improvements have been made in the text but I do not find the major changes that I asked for in the first round of review.
The manuscript has not been revised sufficiently. The analysis of more meaningful samples has not been introduced.
I therefore regret to express my disagreement with the publication of the manuscript.
Reviewer 2 Report
I would have appreciated it more if the authors had re-written some parts of the manuscript based on my comments in the first round of evaluation. However, they have kept almost all the text untouched, even one case, they have just limited to translate some paragraphs from one place to another.
On the other hand, they have, at least, added some details about the used equipment and used methods and images on some captions and they have tried to answer and justify all the posed questions.
I believe that, since the authors have proposed an analysis method and they seem not willing to improve the manuscript if more comments were given, maybe it is useful for some readers of the field, so I propose it to be published in the journal Applied Sciences in its current form.
Author Response
Thank you for your review. We have rewritten and clarified some parts of the article. We hope we have responded better to the reviewer's proposals and comments.
Best regards, Authors